# Absorption of Sulfur Dioxide by Tetraglyme–Sodium Salt Ionic Liquid

**DOI:** 10.3390/molecules24030436

**Published:** 2019-01-26

**Authors:** Qiang Xu, Wei Jiang, Jianbai Xiao, Xionghui Wei

**Affiliations:** College of Chemistry and Molecular Engineering, Peking University, Beijing 100871, China; 1401210206@pku.edu.cn (Q.X.); 1501210153@pku.edu.cn (W.J.); glorioussoviet@126.com (J.X.)

**Keywords:** SO_2_ removal, ionic liquid, recyclable absorption, tetraglyme

## Abstract

A series of tetraglyme–sodium salt ionic liquids have been prepared and found to be promising solvents to absorb SO_2_. The experiments here show that [Na–tetraglyme][SCN] ionic liquid has excellent thermal stability and a 30% increase in SO_2_ absorption capacity compared to other sodium salt ionic liquids and the previously studied lithium salt ionic liquids in terms of molar absorption capacity. The interaction between SO_2_ and the ionic liquid was concluded to be physical absorption by IR and NMR.

## 1. Introduction

SO_2_ (sulfur dioxide) is considered to be one of the inevitable atmospheric pollutants of industrial production. SO_2_ has a number of environmental and health issues and is identified as one of the most important variables to cause hazy weather [1,2]. Thus, the control of SO_2_ emission has been a serious global concern for the last century. As more countries pay extensive attention to environmental protection, limits of SO_2_ emissions in flue gas are becoming more stringent. Among the desulfurization technologies, calcium-based FGD technology is a popular method [3,4,5]. However, the operating cost is an issue. Furthermore, a large amount of gypsum formed in the FGD process has low quality, often clogs the pipeline, and is a secondary pollutant since its permeation is harmful to the soil and groundwater. Therefore, research on the removal of SO_2_ by effective solvents via absorbing-stripping processes has always been a topic of interest. In particular, the development of efficient, low cost, and low volatility solvents is the key to the success of this process.

Organic solvents have recently been used for the removal of SO_2_ [6,7]. However, different organic solvents have their own shortcomings. For example, although organic amines have excellent absorption performance, they are difficult to regenerate. Regarding glycols, almost complete regeneration can be carried out at low temperature, but the absorption capacity is lower than other organic solvents. Glymes were found to have better SO_2_ absorption performance than glycols due to the physical interaction between S atoms and O atoms [8,9]. However, high volatility is one of the major disadvantages of glymes.

Ionic liquids, as a new type of solvent have been further developed for their irreplaceable advantages, such as low saturated vapor pressure and high thermal and chemical stability [10]. Ionic liquids have been used in catalytic processes [11], material synthesis [12], gas detection [13], and separation of mixtures, such as CO_2_ [14], H_2_S [15], NO_2_ [16], and CH_4_ [17] absorption. Different ionic liquids, including guanidinium [18,19,20], imidazolium [21,22,23], hydroxyl ammonium [24,25], pyridinium [26], tetrabutyl ammonium [27], and phosphonium [28] have been used in SO_2_ absorption. These ionic liquids have disadvantages, such as high cost, complexity in preparation, difficulty in regeneration, and so on. In addition, various ether-functionalized [29,30,31] and anion-functionalized [32,33,34] ionic liquids were synthesized to improve the SO_2_ absorption capacity and selectivity. However, most of those ionic liquids are prepared by complicated processes with high cost. This can be a significant obstacle when the ionic liquids’ effective industrial practice is seriously considered. An easily-prepared ionic liquid would have obvious advantages in the industrialization of SO_2_ absorbents.

A series of glyme–lithium salt ionic liquids have been prepared and studied, and the simple synthetic method explored another path for the development of similar ionic liquids [35]. The interaction between Na^+^ and organic solvents, such as crown ether [36], ethylene glycol [37], and acetamide [38], has been extensively studied recently. It was found that sodium salt ionic liquids can be formed by sodium salt dissolution in glymes [39,40].

In this paper, a series of sodium salt ionic liquids were synthesized and the performance of the ionic liquids were studied. It is noticeable that [Na–tetraglyme][SCN] ionic liquid has about 30% better absorption capacity than tetraglyme (G4), other tetraglyme-sodium salt ionic liquids, and the glyme–lithium salt ionic liquids. Moreover, the mechanism of the interaction between SO_2_ and the ionic liquids was investigated by IR and NMR.

## 2. Experimental Section

### 2.1. Preparation of Ionic Liquids

The preparation of the ionic liquid was as follows: first, different sodium salts and tetraglyme were mixed at the stoichiometric ratio of 1:1 and then heated to a temperature of 303 K for 6 h while maintaining sufficient agitation. The solution was then dried in a vacuum drying chamber for 48 h. The resulting solution was transparent yellowish or colorless. The cation of the ionic liquid formed at this time is a supramolecular system consisting of sodium ions and neutral tetraglyme molecules. The anions of the ionic liquids are still the anions initially introduced by the sodium salts. The difference between these ionic liquids is mainly manifested in anions, and the cations have the same structure.

### 2.2. Absorption and Desorption of SO_2_

The absorption experiment was carried out under 1 bar with a SO_2_ partial pressure of 1 bar and a flow rate of 100 mL/min. During the regeneration experiment, regeneration was carried out for 30 min at 80 °C by nitrogen stripping, and the flow rate was 100 mL/min.

## 3. Results and Discussion

### 3.1. Properties of Ionic Liquids

MS, ^1^H-NMR, ^13^C-NMR, and IR were used to identify the structures of the tetraglyme–Na^+^ salt ionic liquids. Low resolution MS data for an ionic liquid consisting of NaSCN and tetraglyme is shown in Appendix A. It can be clearly seen that the [Na–tetraglyme][SCN] ionic liquid has a cationic molecular weight of about 245.1, which is the sum of the molecular weight of tetraglyme (222.28) and the molecular weight of sodium ion (22.99). This means that sodium ion and tetraglyme molecular form the supramolecular structure as the cation of these ionic liquids (shown in Figure 1).

The ^1^H-NMR data of tetraglyme and [Na–tetraglyme][SCN] ionic liquid are shown in Figure 2. After the form of the ionic liquid, the chemical shift has undergone a significant change, which is mainly due to the interaction of the sodium ion with the oxygen atoms in the tetraglyme molecular to reduce the deshielding effect of the oxygen atom. In contrast, the chemical shifts of the carbon atoms are basically unchanged after the form of the ionic liquid (seen in the ^13^C-NMR data in Figure 3). This means that the carbon atoms may not be involved in the interaction with the sodium ions in the formation of the supramolecular structure. The NMR data further confirmed the cationic structure of the ionic liquid (shown in Figure 1).

It can be seen from the IR spectra in Figure 4 that when tetraglyme forms ionic liquid with NaSCN, there is no significant shift in the C–O vibration peak at 1110 cm^−1^ and the C–C vibration peak at 1430 cm^−1^. A closer comparison of the spectra of the two materials reveals another difference: the ionic liquid has a distinct absorption peak at 2064 cm^−1^, which is the absorption peak of SCN^−^.

An important purpose of tetraglyme for forming ionic liquids is to increase the thermal stability of the absorbing solvents. As seen from Figure 5, the thermal stability of [Na–tetraglyme][SCN] ionic liquid is significantly improved compared to tetraglyme. The T_d_ increases from 371 K of tetraglyme to 433 K of [Na–tetraglyme][SCN] ionic liquid. In addition, the mass of [Na–tetraglyme][SCN] ionic liquid remains stable at 373 K, while the tetraglyme is linearly reduced (seen from Appendix A). This means that [Na–tetraglyme][SCN] ionic liquid can be applied to SO_2_ absorbing and regeneration without an obvious solvent loss when operating in the absorption and regeneration temperature range (i.e., 293 K to 353 K), which undoubtedly opens up the possibility of industrial application of those ionic liquids.

### 3.2. Absorption Capacity of Ionic Liquids

Figure 6 shows a comparison of the absorption of SO_2_ by tetraglyme and [Na–tetraglyme][SCN] ionic liquid at different temperatures at one atmosphere. A comparison of the absorption at each temperature indicates a significant increase in the amount of SO_2_ absorbed by the ionic liquid compared to tetraglyme alone. For example, one mol [Na–tetraglyme][SCN] ionic liquid can absorb 2.72 mol SO_2_ at 293 K, while tetraglyme can absorb 2.10 mol SO_2_ under the same conditions. This means the absorption capacity of the ionic liquid is improved by about 30% over tetraglyme. Moreover, the absorption capacity of both tetraglyme and the ionic liquid decreases as the temperature increases, because SO_2_ tends to exist in the form of gas in high temperature.

The absorption capacity of tetraglyme and [Na–tetraglyme][SCN] ionic liquid under different partial pressures of SO_2_ was also investigated and the results are shown in Figure 7. As the partial pressure of SO_2_ increases, the absorption capacity for SO_2_ of both tetraglyme and [Na–tetraglyme][SCN] ionic liquid increases, and this increase is linear. At the same time, the absorption capacity of [Na–tetraglyme][SCN] ionic liquid is always higher than that of tetraglyme alone at the same temperature and partial pressures. 

Figure 8 shows a comparison of the SO_2_ absorption capacity of ionic liquids formed by tetraglyme with several different anionic salts. The saturated SO_2_ absorption of tetraglyme, [Na–tetraglyme][BF_4_] and [Na–tetraglyme][ClO_4_] ionic liquid is 2.10, 2.11, and 2.13 mol per mol solvent, respectively, at the temperature of 293 K and 1 bar. Correspondingly, [Li–tetraglyme][NTf_2_] has a similar SO_2_ absorption capacity to the above ionic liquids of 2.12 mol per mol solvent [35]. As a comparison, the saturated absorption of [Na–tetraglyme][SCN] ionic liquid is 2.72 mol per mol solvent, which is about 30% higher than that of other anionic ionic liquids under the same conditions. It suggests that the type of anion has an important influence on the SO_2_ absorption capacity of the ionic liquid, which is different from previous research [35]. The anion SCN^−^ plays an important role in the absorption of SO_2_. 

In addition, the effects of oxygen and water on the absorption of ionic liquids have also been studied. The results show that oxygen has little effect on the SO_2_ absorption capacity of [Na–tetraglyme][SCN] ionic liquid, as shown in Appendix A. The effect of ionic liquids with different water contents on the absorption capacity of SO_2_ is shown in the Appendix A. As the water content in the ionic liquid increases, the absorption capacity of the ionic liquid increases slightly, rather than decreases. This shows that the participation of water does not hinder the ability of ionic liquids to absorb SO_2_. In summary, the above experiments show that such ionic liquids have good oxygen and water resistance, which is also a required characteristic that can be considered for future industrial desulfurization absorbents.

### 3.3. Regeneration

Tetraglyme and [Na–tetraglyme][SCN] ionic liquid were also used in absorption and regeneration tests, as shown in Figure 9. The result shows that regardless of tetraglyme and the ionic liquid, the solvents maintain good absorption and regeneration performance, and the SO_2_ absorption capacity of the ionic liquid is always higher than that of tetraglyme by about 30%. In all the five cycles, both tetraglyme and the ionic liquid have a low sulfur dioxide content after the regeneration. The difference of SO_2_ content in the solvents in the absorbing/regenerating cycles can lead to the conclusion that [Na–tetraglyme][SCN] ionic liquid is more effective for future applications than tetraglyme.

### 3.4. Mechanism

IR, ^1^H-NMR, and ^13^C-NMR were applied to further investigate the interaction between these ionic liquids and SO_2_. The IR spectral result shows that no new peak appeared after the absorption of SO_2_ (seen in Figure 4), i.e., no new chemical bond is formed. This means that there is no chemical interaction between SO_2_ and the sodium salt ionic liquid, which is consistent with the previous experimental results and is consistent with the conclusions from the literature on the absorption of tetraglyme [9]. That is to say, the formation of ionic liquids does not fundamentally change the nature of absorption.

The ^1^H-NMR result in Figure 2 shows that the chemical shift changes of tetraglyme and [Na–tetraglyme][SCN] ionic liquid are similar after absorption of SO_2_, which means that tetraglyme and [Na–tetraglyme][SCN] ionic liquid have the similar mechanism of interaction between the solvents and SO_2_, based on the charge-transfer interaction between sulfur atoms in SO_2_ and oxygen atoms in tetraglyme. The ^13^C-NMR result in Figure 3 further confirms this mechanism. Only the chemical shift belonging to the carbon atoms in SCN^−^ moves after SO_2_ absorption. This means there is no obvious interaction between the carbon atoms in tetraglyme and SO_2_. However, the carbon atom in SCN^−^ moves upfield after the absorption of SO_2_, which suggests that there is Van der Waals’ force between SO_2_ and SCN^−^. In other words, SCN^−^ plays a significant role in the SO_2_ absorption process of the [Na–tetraglyme][SCN] ionic liquid. 

## 4. Conclusions

In conclusion, a series of tetraglyme–sodium salt ionic liquids were prepared. Their structures were characterized, and their absorption capacities of SO_2_ were tested. In addition, the interaction between SO_2_ and the ionic liquids were investigated. The formed [Na–tetraglyme]^+^ can not only significantly improve the thermal stability of the solvent but also effectively reduce solvent volatilization. It is clear that the ionic liquids mentioned above, especially [Na–tetraglyme][SCN] ionic liquid, are excellent SO_2_ absorbents, considering both the good absorption and impressive regeneration performance. Moreover, ^1^H-NMR, ^13^C-NMR, and IR were applied to analyze the absorption mechanism of SO_2_ absorption in these ionic liquids. The results suggest that charge-transfer interaction between sulfur atoms and oxygen atoms is the main force, and the Van der Waals’ force between SCN^−^ and SO_2_ also plays an important role. As a conclusion, tetraglyme–sodium salt ionic liquids with high thermal stability and excellent SO_2_ absorption capacity can be picked out as promising alternatives to the traditional SO_2_-absorbing agents in SO_2_ removal. 

## Figures and Tables

**Figure 1 molecules-24-00436-f001:**
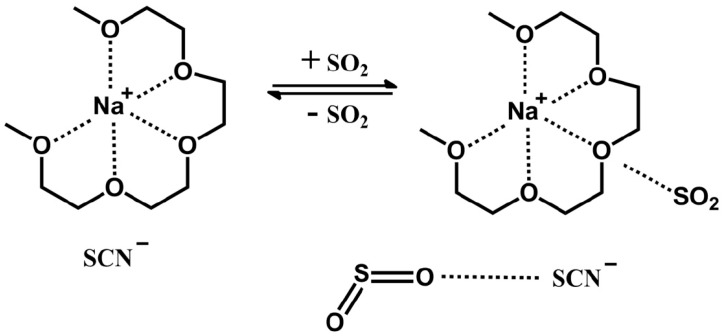
Structure of [Na–tetraglyme]^+^ ion before and after SO_2_ absorption.

**Figure 2 molecules-24-00436-f002:**
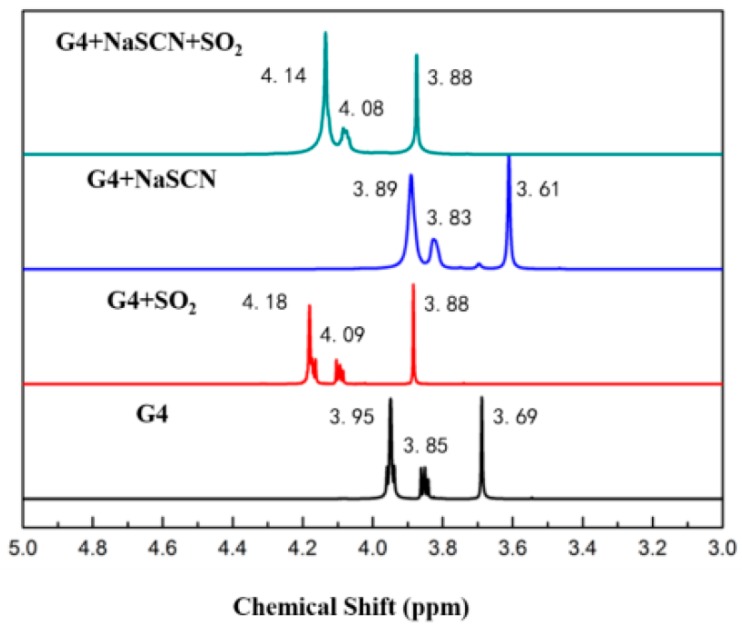
^1^H-NMR spectra of tetraglyme, tetraglyme after SO_2_ absorption, [Na–tetraglyme][SCN] and [Na–tetraglyme][SCN] after SO_2_ absorption, with CDCl_3_ as an external reference.

**Figure 3 molecules-24-00436-f003:**
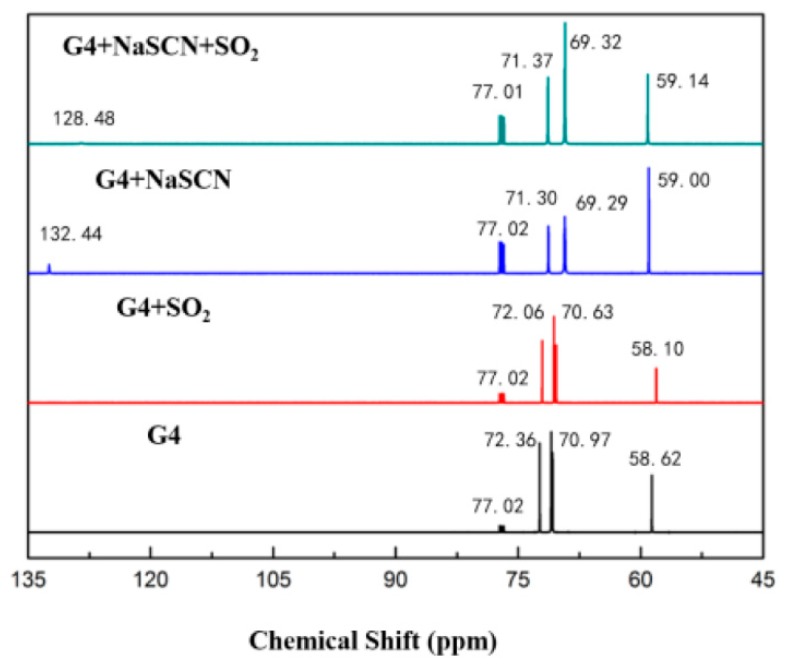
^13^C-NMR spectra of tetraglyme, tetraglyme after SO_2_ absorption, [Na–tetraglyme][SCN] and [Na–tetraglyme][SCN] after SO_2_ absorption, with CDCl_3_ as an external reference.

**Figure 4 molecules-24-00436-f004:**
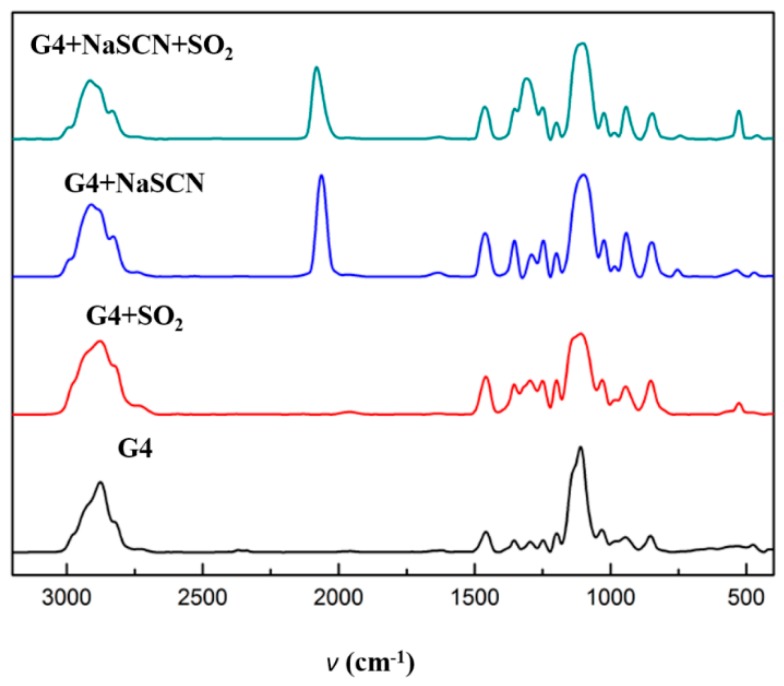
IR spectra of tetraglyme and [Na–tetraglyme][SCN] before and after SO_2_ absorption.

**Figure 5 molecules-24-00436-f005:**
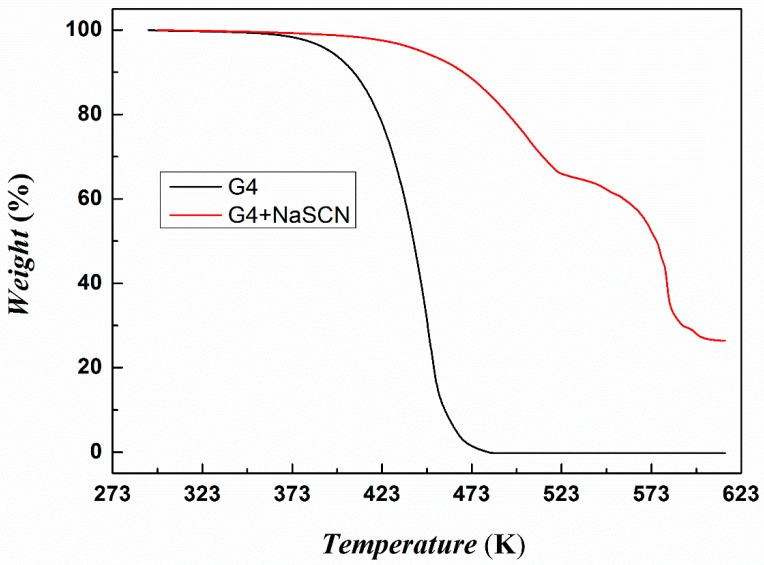
Thermal gravimetric analysis of tetraglyme and [Na–tetraglyme][SCN].

**Figure 6 molecules-24-00436-f006:**
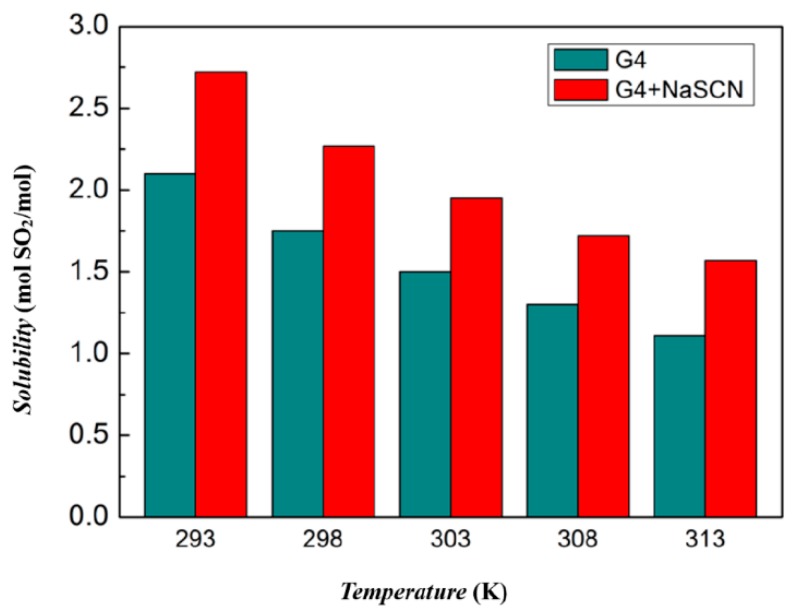
SO_2_ absorption capacities of tetraglyme and [Na–tetraglyme][SCN] at different temperatures with the pressure of SO_2_ equal to 1 bar.

**Figure 7 molecules-24-00436-f007:**
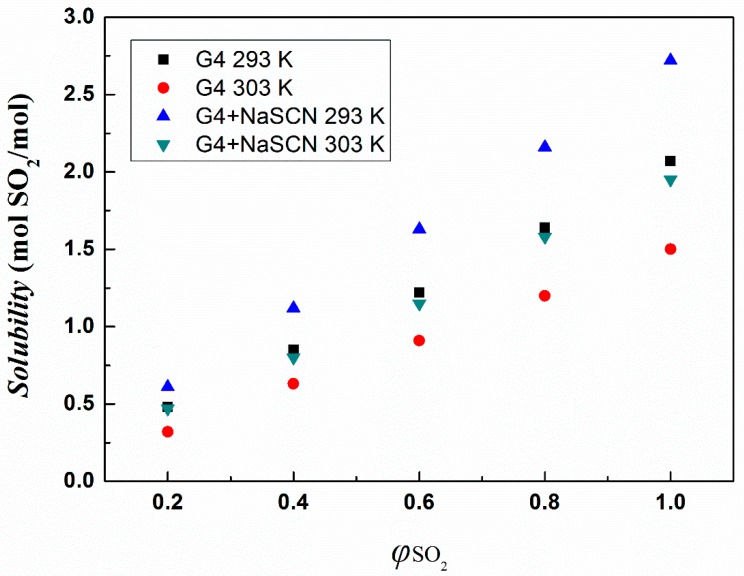
SO_2_ absorption capacities of tetraglyme and [Na–tetraglyme][SCN] at different SO_2_ partial pressures.

**Figure 8 molecules-24-00436-f008:**
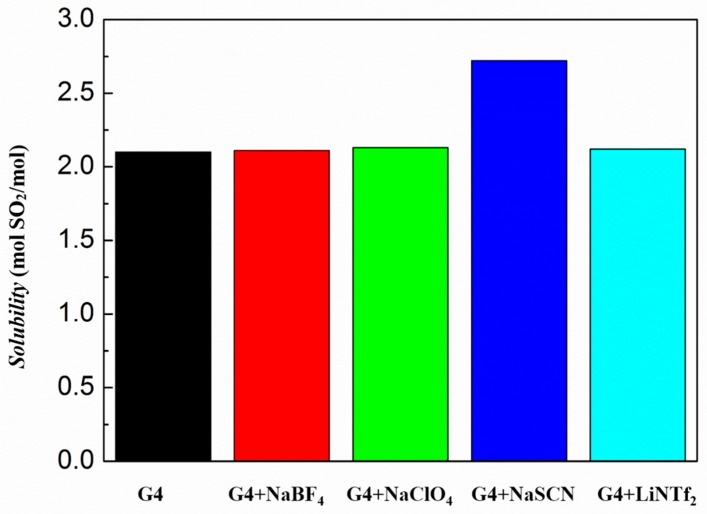
SO_2_ absorption capacities of tetraglyme, [Li–tetraglyme][NTf_2_] ionic liquid, and [Na–tetraglyme][X] ionic liquids at 293 K and 1 bar (X = BF_4_, ClO_4_, and SCN).

**Figure 9 molecules-24-00436-f009:**
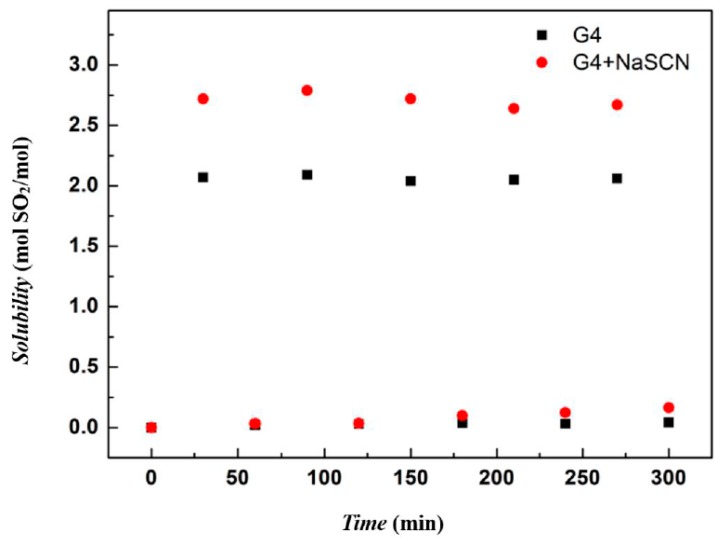
SO_2_ absorption capacities of tetraglyme and [Na–tetraglyme][SCN] over five absorption–desorption cycles.

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
