# Peer review of "Absorption of Sulfur Dioxide by Tetraglyme–Sodium Salt Ionic Liquid"

_molecules, 2019, doi:10.3390/molecules24030436_

Round 1

Reviewer 1 Report

The manuscript by Chang Xu al (molecules-412434) focuses on the synthesis and characterization of a series of tetraglyme-sodium salt ILs and their SO2 absorption properties. This manuscript can be accepted after addressing the following comments.

1.       Abstract talks about Lithium ILs. But no data or literature comparison in the main text.

2.       Authors should provide data and justification for interferences such as Oxygen for the SO2 absorption in atmospheric conditions.

3.       Is this ILs moisture sensitive? Please comment.

4.       Explain the necessity of 48 hours vacuum drying in synthesis steps.

5.       Please specify the chemical purity of Tetraglyme in supporting information section.

6.       Please provide the percent yields for each IL synthesis.

7.       Specify whether the MS confirmation is high resolution or not.

8.       Line 76: Define ESM

9.       Please check Figure 2 and 3 captions, “spectraon” typo? Also, what is inside brackets is not clear.

10.   Figure 5: temperature units in Celsius. Please be consistent with the text.

11.   Line124: Explain why absorption capacity decreases with increasing temperature.

Author Response

Thank you for your review. We revised the manuscript according to your reply and corrected most of the grammatical errors. 

1. Abstract talks about Lithium ILs. But no data or literature comparison in the main text.

Respond: Thank you for your kind suggestion. We added absorption data of Lithium IL in the manuscript.

2. Authors should provide data and justification for interferences such as Oxygen for the SO2 absorption in atmospheric conditions.

Respond: Thank you for your kind suggestion. We carried out SO2 absorption experiments in different SO2 partial pressure with air as equilibrium gas in order to judge the influence of O2 in the absorption process. The results are shown in the manuscript and the figure is shown in supporting information. The results show that oxygen has little effect on the SO2 absorption capacity of [Na-tetraglyme][SCN] ionic liquid.

3. Is this ILs moisture sensitive? Please comment.

Respond: Thank you for your kind suggestion. We carried out SO2 absorption experiments by IL with different water content in order to judge the influence of water in the absorption process. The results are shown in the manuscript and the figure is shown in supporting information. The results show that water has no adverse effect on the SO2 absorption capacity of [Na-tetraglyme][SCN] ionic liquid.

4. Explain the necessity of 48 hours vacuum drying in synthesis steps.

Respond: Thank you for your kind question. Supplementary experiments have shown that water has no adverse effect on the absorption of SO2 by ionic liquids. The main purpose of this step is to remove water and volatile organic compounds from ionic liquids, so as to avoid the interference of these substances in the study of experimental mechanism.

5. Please specify the chemical purity of Tetraglyme in supporting information section.

Respond: Thank you for your kind suggestion. We confirm that the chemical purity of Tetraglyme is 99% and we have added it to the support information.

6. Please provide the percent yields for each IL synthesis.

Respond: Thank you for your kind question. However, the ionic liquids in this paper have a similar property to protic ionic liquids. These ionic liquids have anions or cations complex structure and there are interactions other than chemical bonds such as ion-dipole interaction and Van der Waals' force. This means that the percent yields of these ionic liquids cannot be determined.

7. Specify whether the MS confirmation is high resolution or not.

Respond: Thank you for your kind suggestion. Low resolution MS was used and we revised it in the manuscript.

8. Line 76: Define ESM.

Respond: Thank you for pointing this error. We corrected it in the manuscript.

9. Please check Figure 2 and 3 captions, “spectraon” typo? Also, what is inside brackets is not clear.

Respond: Thank you for pointing this error. We corrected it in the manuscript.

10. Figure 5: temperature units in Celsius. Please be consistent with the text.

Respond: Thank you for pointing this error. We corrected it in the manuscript.

11. Line124: Explain why absorption capacity decreases with increasing temperature.

Respond: Thank you for your kind question. It is known that the solubility of most gases in liquids decreases with increasing temperature. With the increase of temperature, SO2 tends to exist in the form of gas, so that absorption capacity decreases with increasing temperature.

Reviewer 2 Report

I do not recommend publishing in this form for publication.

1. The interaction of tetraglyme with sodium ion or SO2 has been known for many years. The publication does not bring any scientific novelty in this area.

2. Glymes as substances reacting with SO2 and releasing this compound after changing external conditions (temperatures) are used in many sulfuric acid factories for the purification of flue gases.

3. There is a publication describing an identical process for the Li salt.

Highly efficient sulfur dioxide capture by glyme-lithium salt ionic liquids (DOI: 10.1039/x0xx00000x)

(A series of glyme-lithium salt ionic liquids were prepared and applied in SO2 absorption. The formed quasi-aza-crown ether fashions between Li+and glymes can effectively reduce the solvent volatilization, and have an excellent SO2 absorption capacity. In addition, the mechanism of interaction between SO2 and ionic liquids was investigated by IR and NMR.)

The conversion of lithium salt to sodium salts did not cause any changes in the mechanism of process. No discussion in the publication of the purpose of converting salt in this process.

4. There is no data on the equilibrium constant of complexation process and reaction with gases. What is the difference in the values of  equilibrium constant reaction of games with SO2 and the tested complex with SO2?

5. The procedure for obtaining the salt complex from glymes does not provide existed of free glyme. Experimentally, the formation of a salt anion complex with SO2 has not been documented.

6. What is the purpose of this process (The solution was then dried in a vacuum drying chamber for 48 hours. The resulting solution is transparent yellowish or colorless) in the synthesis of the complex when pure reagents were used?

Author Response

Thank you for your review. We revised the manuscript according to your reply and corrected most of the grammatical errors.

1. The interaction of tetraglyme with sodium ion or SO2 has been known for many years. The publication does not bring any scientific novelty in this area.

2. Glymes as substances reacting with SO2 and releasing this compound after changing external conditions (temperatures) are used in many sulfuric acid factories for the purification of flue gases.

3. There is a publication describing an identical process for the Li salt.

Highly efficient sulfur dioxide capture by glyme-lithium salt ionic liquids (DOI: 10.1039/x0xx00000x)

(A series of glyme-lithium salt ionic liquids were prepared and applied in SO2 absorption. The formed quasi-aza-crown ether fashions between Li+ and glymes can effectively reduce the solvent volatilization, and have an excellent SO2 absorption capacity. In addition, the mechanism of interaction between SO2 and ionic liquids was investigated by IR and NMR.)

The conversion of lithium salt to sodium salts did not cause any changes in the mechanism of process. No discussion in the publication of the purpose of converting salt in this process

Respond1&2&3: Thank you for this kind suggestion. [Na-tetraglyme][SCN] ionic liquid in this paper have a better SO2 absorption capacity than tetraglyme and those Li salt ILs. And this ILs have better thermal stability than glymes. Moreover, the cost of these ILs is acceptable.

4. There is no data on the equilibrium constant of complexation process and reaction with gases. What is the difference in the values of equilibrium constant reaction of games with SO2 and the tested complex with SO2?

Respond: Good questions! Researchers should pay more attentions on the study of this kind of ILs and more characterization method should be used on these study. However, there is no effective method to confirm the equilibrium constant experimentally. We have known there is sure ILs prepared through the MS results.

5. The procedure for obtaining the salt complex from glymes does not provide existed of free glyme. Experimentally, the formation of a salt anion complex with SO2 has not been documented.

Respond: Thank you for this kind question. Actually, the ionic liquids in this paper have a similar property to protic ionic liquids. These ionic liquids have anions or cations complex structure and there are interactions other than chemical bonds such as ion-dipole interaction and Van der Waals' force. Ideally, Na+ reacts perfectly with G4, so that the only single species present are the cations and anions produced. In reality, the reaction could not fully complete and there should be free G4. But there is currently no “standard” method for determining the ionicity of these ILs nor a standard method for classifying which should be considered pure ionic liquids.

6. What is the purpose of this process (The solution was then dried in a vacuum drying chamber for 48 hours. The resulting solution is transparent yellowish or colorless) in the synthesis of the complex when pure reagents were used?

Respond: Thank you for this kind question. The main purpose of this step is to remove water and volatile organic compounds from ionic liquids, so as to avoid the interference of these substances in the study of experimental mechanism. In addition, we carried out SO2 absorption experiments by IL with different water content in order to judge the influence of water in the absorption process. The results are shown in the manuscript and the figure is shown in supporting information. The results show that water has no adverse effect on the SO2 absorption capacity of [Na-tetraglyme][SCN] ionic liquid.

Reviewer 3 Report

This article highlights SO2 absorption using ionic liquid. This is nicely written and described in a nice fashion. This article could be accepted.

Author Response

Thank you for your kind review. We corrected most of the grammatical errors in the manuscript.

Round 2

Reviewer 1 Report

The following minor changes are recommended before accepting the manuscript.

Line 36: Introduction, briefly explain the use of ILs for other type of extraction platforms with literature citations (Ex: ACS applied materials & interfaces 9 (29), 24955-24963)

Line 48: I would not start the sentence with “And…”

Line 122: Please check the font. Seems it is different to the rest

Line 128: SO2 instead sulfur dioxide

Line 138: If Li ILs are previously studied and published, please include the citation

Line 238: Please check the journal abbreviation

Line 243: what is “……”?

Specify dimensions of absorption tube (Ex: length) in SI section and how much IL in the tube.

Author Response

1. Line 36: Introduction, briefly explain the use of ILs for other type of extraction platforms with literature citations (Ex: ACS applied materials & interfaces 9 (29), 24955-24963).

Respond: Thank you for your kind suggestion. We added literatures on ionic liquid applications and briefly introduced the absorption of other gases by ionic liquids.

2. Line 48: I would not start the sentence with “And…”

Respond: Thank you for your kind suggestion. We deleted “And” here.

3. Line 122: Please check the font. Seems it is different to the rest.

Respond: Thank you for pointing this error. We corrected this error in the manuscript.

4. Line 128: SO2 instead sulfur dioxide.

Respond: Thank you for pointing this error. We corrected this error in the manuscript.

5. Line 138: If Li ILs are previously studied and published, please include the citation.

Respond: Thank you for your kind suggestion. We added the citation here. The data is slightly different in this manuscript caused by the difference of SO2 flow rate.

6. Line 238: Please check the journal abbreviation.

Respond: Thank you for pointing this error. We corrected this error in the manuscript.

7. Line 243: what is “……”?

Respond: Thank you for pointing this error. We corrected this error in the manuscript.

8. Specify dimensions of absorption tube (Ex: length) in SI section and how much IL in the tube.

Respond: Thank you for your kind suggestions. We added the length and the dosage of ILs in SI section.

Reviewer 2 Report

The authors of the publication presented their view on the basic questions asked in the review without supporting them with the results of the research. In this approach, the publication describes the phenomenon without explaining its mechanism. NMR and FTIR studies of materials would significantly broaden the description of the process. The described processes only broaden the already known knowledge to a small extent and may be the basis for further research.

As a work signaling the possibility of using ethers to bind sulfur oxides, I recommend working for publication.

Author Response

Thank you for your kind review.